# Advances in the Biosynthesis of Plant Terpenoids: Models, Mechanisms, and Applications

**DOI:** 10.3390/plants14101428

**Published:** 2025-05-10

**Authors:** Renwu Cheng, Shuqi Yang, Dongli Wang, Fangcuo Qin, Shengkun Wang, Sen Meng

**Affiliations:** 1Guangzhou Collaborative Innovation Center on Science-Tech of Ecology and Landscape, Guangzhou 510520, China; crw_cheng@163.com; 2State Key Laboratory of Tree Genetics and Breeding, Research Institute of Tropical Forestry, Chinese Academy of Forestry, Guangzhou 510520, China; yshuqi2021@163.com (S.Y.); qinfc@caf.ac.cn (F.Q.); wskun2001@163.com (S.W.); 3College of Agriculture, Guangxi University, Nanning 530004, China; wangdongli1997@163.com; 4Jiangxi Provincial Key Laboratory of Plantation and High Valued Utilization of Specialty Fruit Tree and Tea, Institute of Biological Resources, Jiangxi Academy of Sciences, Nanchang 330096, China

**Keywords:** essential oil, mevalonate pathway, methylerythritol phosphate pathway, secondary metabolites, VOCs

## Abstract

Plants have evolved complex terpene defenses. Terpenoids accumulate in plant tissues or release as volatile in response to ever-changing environment, playing essential roles in chemo-ecological functions as defense against pathogen and insect, improving pollination and seed dispersal, facilitation plant-to-plant communication. They are also gaining attention in pharmaceuticals, nutraceuticals, fragrance, and biofuels. Here, we highlight the recent progress in the fundamental pathways of terpenoid biosynthesis, key enzymes, and their corresponding genes involved in terpenoid synthesis. We identified the further exploration of biosynthetic networks and the development of novel terpenoid resources, proposed the need for further exploration of biosynthetic networks and the development of novel terpenoid resources. Based on that knowledge, future research should be directed towards the mechanisms governing terpenoid biosynthesis dependent environmental change and molecular breeding.

## 1. Introduction

Terpenoids constitute a significant proportion of plant secondary metabolites [1]. These compounds are derived from five-carbon precursors, commonly referred to as isoprene units, which have a propensity to decompose into isoprene (C_5_H_8_) at elevated temperatures [2,3]. Isoprene units assemble to yield a hierarchical series of terpenes, ranging from 5-carbon hemiterpenes to 40-carbon tetraterpenes, including monoterpenes (C_10_H_16_), sesquiterpenes (C_15_H_24_), diterpenes (C_20_H_32_), sesterterpenes (C_25_H_40_), and triterpenes (C_30_H_48_) [4].

For a long time, they were mistakenly regarded as mere metabolic waste or detoxification byproducts. Since the 1970s, scientists have revealed that many terpenoids exhibit bioactive properties, functioning as toxins, repellents, or attractants to other organisms [5]. In recent years, the biological significance of secondary metabolites, including terpenes and isoprenoids, is also gaining attention for their applications in pharmaceuticals, nutraceuticals, synthetic chemistry, the flavor and fragrance industries, and biofuels [6]. In addition, with the escalating demand for these natural products, synthetic biology has emerged as a transformative approach, offering innovative strategies to optimize the production of these bioactive compounds.

This review systematically summarizes the biochemical pathways of terpenoid biosynthesis in plant secondary metabolism, elucidating their synthetic principles from the perspective of metabolic network regulation. Through a comparative analysis of the molecular mechanisms underlying different terpenoid synthesis pathways, we reveal the foundational basis of structural diversity. Furthermore, we highlight recent advances in molecular regulation, including transcriptional control, miRNA-mediated modifications, and epigenetic regulation. These systematic insights not only provide a theoretical framework for deciphering unknown terpenoid biosynthesis mechanisms but also establish a critical foundation for the rational design and efficient production of terpenoids via synthetic biology approaches. The findings hold significant theoretical and practical value for further unraveling the regulatory logic of plant terpenoid metabolism and developing high-value terpenoid products.

## 2. Model of Terpenoid Biosynthesis

Terpene precursors, isopentenyl diphosphate (IPP) and dimethylallyl diphosphate (DMAPP), are synthesized through two distinct pathways: the mevalonate pathway (MVA pathway) and the 2C-methyl-D-erythritol-4-phosphate pathway (MEP pathway). These pathways are localized in different subcellular compartments and utilize different starting materials, providing a fundamental carbon skeleton for the synthesis of a wide variety of terpenoid compounds [7]. While the MEP pathway utilizes pyruvate and D-glyceraldehyde-3-phosphate (GAP), two central metabolites in glycolysis, the MVA pathway begins with acetyl-CoA, generated via the decarboxylation of pyruvate, another pivotal glycolytic product [8,9]. Isoprenyl diphosphate synthases (IDSs) subsequently catalyze the formation of geranyl diphosphate (GPP), farnesyl diphosphate (FPP) [10], and geranylgeranyl diphosphate (GGPP), which serve as the precursors for terpenoid biosynthesis. These intermediates are then processed by terpene synthase (TPS) enzymes, which generate a structurally diverse array of terpenes through stereospecific cyclization and rearrangement reactions [11]. Further structural elaboration occurs via oxidative modifications mediated by cytochrome P450 oxygenases (CYP450s), which modulate the functional diversity of terpenoids [12] (Figure 1).

### 2.1. The MVA and MEP Pathways Generate Initial Precursors

The MVA pathway (Figure 2), a specialized metabolic network predominantly localized to the cytoplasm and endoplasmic reticulum (ER), with potential peroxisomal contributions [13], orchestrates the synthesis of IPP through a six-enzyme cascade. This process consumes three acetyl-CoA molecules, three ATP equivalents, and two NADPH molecules to yield a single IPP molecule [7]. A pivotal rate-limiting step occurs at the conversion of HMG-CoA to mevalonate, catalyzed by HMG-CoA reductase (HMGR) within the ER membrane. This reaction uniquely consumes two NADPH molecules, underscoring its regulatory significance [14]. The final step—conversion of mevalonate-5-diphosphate to IPP by mevalonate diphosphate decarboxylase (MVD)—is hypothesized to occur in peroxisomes [15,16], although cytoplasmic localization remains debated. It is notably that the IPP product undergoes reversible isomerization to DMAPP via IPPI, a critical reaction for downstream terpenoid biosynthesis [17].

The metabolic sequence of the MEP pathway occurs exclusively within the plastid and involves a series of seven reactions that convert GAP and pyruvate into isopentenyl diphosphate (IDP) and DMADP (Figure 3). This conversion involves condensation and reduction processes, with the consumption of three ATP and three NADPH molecules [7]. Initially, 1-deoxy-D-xylulose-5-phosphate synthase (DXS) catalyzes the condensation of pyruvate and GAP to form 1-deoxy-D-xylulose-5-phosphate (DXP). This step is crucial in the MEP pathway, as DXS activity significantly influences the metabolic rate of the entire pathway [18]. Additionally, the MEP pathway provides essential precursors for the biosynthesis of carotenoids and chlorophyll alcohols in chloroplasts, which are critical components of photosynthesis and other primary metabolic processes [19].

In plants, isoprene biosynthesis uniquely involves both the MVA pathway and the MEP pathway, contrasting with the single-pathway utilization observed in other organisms such as animals, fungi, and most bacteria [9]. This dual-pathway approach in plants is an evolutionary adaptation that provides significant survival advantages [20]. The compartmentalization of these pathways in distinct cellular locations (the MVA pathway predominantly in the cytoplasm/endoplasmic reticulum and the MEP pathway within the plastids) allows the efficient utilization of diverse carbon sources. This strategic localization facilitates the rapid and specific production of end products, reducing substrate competition and enhancing enzyme efficiency [21]. Furthermore, the presence of both pathways provides plants with the flexibility to adapt to various environmental conditions. It has been observed that, under light conditions, the MEP pathway becomes more active, supporting the synthesis of photosynthesis-related isoprenoids. In contrast, in the absence of light, the MVA pathway exhibits heightened activity, promoting the synthesis of phytosterols [9]. Despite these observations, the precise regulatory mechanisms governing the coordinated interaction between the MVA and MEP pathways remain unclear. Further research is needed to elucidate the complex regulatory networks that ensure the efficient coordination of these pathways in response to environmental change.

Subcellular localization studies of IPPI have revealed its presence in mitochondria, peroxisomes, and chloroplasts [22]. This indicated a potential transfer mechanism of IPPI across these distinct cellular compartments. Various experimental methodologies have been applied to confirm the existence of cross-pathway flow between the MVA and MEP pathways, including mutant analyses, chemical inhibitor interventions, and labeled substrate feeding experiments [23]. Skorupinska-Tudek et al. [24] utilized nuclear magnetic resonance (NMR) and mass spectrometry to observe frequent exchanges between plastids and the cytoplasm, leading to the formation of compounds with mixed MVA/MEP origins. Lipko et al. [25] employed a novel competitive labeling approach, providing additional evidence for the intricate interconnection and reciprocal regulation between the MEP and MVA pathways. Despite these findings, the extent of this metabolic flux remains limited and tightly regulated, the precise regulatory mechanisms governing this interaction continue to be an area of active investigation.

### 2.2. Direct Precursor Formation Stage

All terpenoids are derived from DMAPP and IPP through the enzymatic action of various IDSs. These enzymes catalyze the sequential condensation of isoprenoid units by ionizing the allylic diphosphate ester, followed by a rearrangement that results in the formation of prenyl diphosphate metabolites of varying chain lengths [26,27]. The initial condensation of one DMAPP and one IPP, catalyzed by GPPS, yields GPP, the precursor of monoterpenes. GPP synthesis primarily occurs in plastids and mitochondria [28]. Further elongation via FPPS incorporates an additional IPP with GPP to form FPP, serving as the backbone of sesquiterpenes [29]. GGPPS catalyzes the addition of three IPPs to one DMAPP, generating GGPP, a precursor for diterpenes that localizes to the cytoplasm, plastids, and mitochondria [28]. FPP and GGPP can then undergo further polymerization to form triterpenes and tetraterpenes, respectively [30] (Figure 1).

The structural diversity of terpenoids begins to diverge at the level of IDS enzymes, manifested through variations in enzyme architecture and alternative catalytic mechanisms that generate unconventional terpene scaffolds. Notably, direct precursors for terpenoid biosynthesis are initially in the trans (E) configuration and undergo “head-to-tail” condensation reactions to form terpenoids [31,32]. Cis-isoprenoid diphosphate synthases (cis-IDS) exhibit remarkable structural diversity in their active sites and broad functional capacities, unlike their trans-isoprenoid counterparts (trans-IDS). These enzymes are referred to as “butterfly-fold” proteins due to their characteristic structural resemblance to a spreading butterfly [33]. Research on the trichomes of cultivated tomato (*Solanum lycopersicum*) has identified three cis-configured intermediates: NPP, Z,Z-FPP, and NNPP. NPP is synthesized by NPPS1/S1CPT1; Z,Z-FPP is produced by zFPPS; and NNPP is generated by NNPPS/S1CPT2 [34]. The structural composition of the subunits of IDS determines whether they form homodimers or heterodimers. A cis-heterodimer IDS contains a non-catalytic subunit that facilitates the production of long-chain polyprenyl diphosphates. For instance, natural rubber (C_10,000_), the largest naturally occurring hydrocarbon molecule, is a long-chain Z-type isoprenoid produced through such mechanisms [26]. “Non-head-to-tail” condensation of two DMAPP molecules can produce branched structures. Rivera et al. [35] isolated a monoterpene with an irregular C1′-2-3 linkage between isoprenoid units, termed CPP, from *Chrysanthemum cinerariaefolium*, which was catalyzed by CPPS. Demissie et al. [36] also identified a novel cis-prenyl diphosphate synthase cDNA from *Lavandula x intermedia*, designated LiLPPS. This enzyme, LPPS, catalyzes the non-head-to-tail condensation of isoprene units to form LPP (Figure 1).

### 2.3. TPS Catalyzes the Subsequent Steps

The fundamental structural framework of terpenes is established by TPS [4]. The TPS gene family represents a moderate-sized group in plants, hypothesized to have originated from an ancestral bifunctional gene encoding an enzyme with both copalyl diphosphate synthase (CPS) and kaurene synthase (KS) activities. Over time, this gene diverged into two separate paralogs—one retaining CPS activity and the other KS activity—which became the basis for different TPS subfamilies [37]. The diversification of TPS subfamilies occurred after land plants split from *Charophytic algae*, driven by repeated gene duplications that led to the evolution of distinct subfamilies [38]. TPS enzymes exhibit remarkable regioselectivity in substrate binding, enabling them to utilize diverse precursors like GPP, FPP, and GGPP to generate a broad spectrum of terpenoid products [39]. Additionally, the conformation of a product can be dictated by TPS enzymes. For instance, in Mentha canadensis, a single TPS catalyzes the formation of both L-menthol and D-menthol [5].

The evolutionary relationships among these TPS classes and subfamilies are illustrated in Figure 4. TTS and PT, sharing a conserved α-domain with a DDx2D motif similar to ancestral IDS enzymes, evolved through gene fusion (forming αβγ domains) and subsequent domain loss/specialization, giving rise to modern plant Class I/II TPS and smaller TPS families. The TPS gene family is divided into seven subfamilies (TPS-a, b, c, d, e/f, g, and h) based on sequence and function [40]. TPS-a mainly synthesizes sesquiterpenes, while TPS-b produces monoterpenes. TPS-g, closely related to TPS-b, forms acyclic mono-, sesqui-, and diterpenes [38]. TPS-c is crucial for diterpenoid biosynthesis in plant secondary metabolism [41]. TPS-d acts as both mono- and diterpene synthase. TPS-e/f (evolutionarily linked) catalyzes diterpene synthesis, while TPS-h, unique to gymnosperms, generates mono- and sesquiterpenes [42]. TPS enzymes can be categorized into two primary groups based on their distinct catalytic mechanisms: Class I and Class II. Class I TPSs use an ionization-dependent mechanism. They contain two conserved domains—the C-terminal DDxxD motif and the (L, V) (V, L, A)—(N, D) D (L, I, V) × (S, T) xxxE domain (NSE/DTE domain)—which coordinate metal ions and stabilize carbocation intermediates. Class II TPSs employ a proton-dependent mechanism and feature the DxDD conserved domain [30,42].

### 2.4. Modification of the Forming Terpenoids

After the synthesis of the terpenoid carbon skeleton, further structural diversification and functional modification are facilitated by CYP450 enzymes [4]. “P450” originates from the characteristic absorption band at 450 nm, observed when carbon monoxide binds to the reduced form of the heme iron in these enzymes [43]. Cytochrome P450 represents one of the largest and most ancient gene superfamilies, found across all living organisms. These enzymes catalyze a diverse range of chemical reactions [12]. In mint (*Mentha* spp.), the CYP450 enzyme limonene 3-hydroxylase specifically catalyzes the regiospecific hydroxylation of limonene at the C3 position, yielding trans-isopiperitenol [44]. In *Artemisia annua*, CYP71AV1 mediates the sequential oxidation of key intermediates in artemisinin biosynthesis, including amorpha-4,11-diene, artemisinol, and artemisinic aldehyde [45]. In *Tripterygium wilfordii*, members of the CYP71BE subfamily have been shown to catalyze the 18(4→3) methyl transfer reaction, a key step in the biosynthesis of diterpenoid triepoxides and the bicyclic sesquiterpene core structures found in numerous plant diterpenes [46].

Specific CYP450 proteins have been identified to catalyze the formation of homoterpenes—terpenoids with irregular carbon skeletons [4]. Two prominent examples are (E)-4,8-dimethyl-1,3,7-nonatriene (DMNT) and (E,E)-4,8,12-trimethyltrideca-1,3,7,11-tetraene (TMTT), both widespread in plants. These compounds are typically found in the flowers or are released in response to herbivore feeding [47,48]. Stable isotope labeling studies show that DMNT biosynthesis starts from FDP, which is first converted to the intermediate (E)-nerolidol and then oxidatively degraded to form DMNT [49]. Similarly, GGPP is transformed into (E,E)-geranyllinalool before oxidative cleavage produces TMTT [50]. In *Arabidopsis thaliana*, the biosynthetic pathway of the volatile sesquiterpene TMTT has been fully characterized. This pathway involves the CYP82G1 enzyme, which specifically catalyzes the oxidative conversion of (E,E)-geranyllinalool to form TMTT [47,51].

## 3. Regulation of Terpene Biosynthesis

### 3.1. Transcription Factors in Terpene Biosynthesis

Transcription factors (TFs) are crucial regulatory proteins in plants that modulate gene expression by binding to cis-regulatory elements, either proximal or distal to their target genes. In total, 64 distinct transcription factor families have been identified in vascular plants. Among these, several families play pivotal roles in regulating plant secondary metabolism, including the following: WRKY, myeloblastosis-related (MYB), NAC, APETALA2/ethylene response factor (AP2/ERF), basic helix–loop–helix (bHLH), basic leucine zipper (bZIP), and others [52]. Kelimujiang et al. [53] identified 12 *LaWRKY* genes in *Lavandula angustifolia* that exhibit elevated expression levels in the buds and calyx, which are the primary organs of volatile terpenoid production. In tomato, the inhibition of SlMYB75 led to an increased accumulation of sesquiterpenes such as δ-elemene, β-caryophyllene, and α-humulene [54]. Li et al. [55] found that SaNAC30 in sandalwood (*Santalum album*) interacts with MKS during terpenoid biosynthesis in sandalwood heartwood. Meng et al. [56] observed a significant increase in *SaAREB6* expression in sandalwood under drought conditions, accompanied by a concurrent increase in santalol content. Yi and colleagues [57] identified 12 TPS-related *AarbHLH* genes in *Artemisia argyi* that may be involved in the synthesis of 1,8-cineole or β-caryophyllene. Wang et al. [58] performed RNA-Seq analysis of the leaf, root, and stem tissues in *Polygonatum kingianum*, identifying 12 *PkARF* genes associated with metabolic pathways and secondary metabolite biosynthesis in roots.

### 3.2. MiRNAs in Terpene Biosynthesis

In addition to TFs, the involvement of microRNAs (miRNAs) in the regulation of terpenoid biosynthesis has been increasingly reported in recent years. MiRNAs regulate critical enzymes involved in both the MVA and MEP pathways, as well as downstream IDS and TPS. In *Catharanthus roseus*, miR-5021 has been identified as targeting HDS [59]. Tu et al. [60] conducted a multi-omics analysis to investigate miRNAs in *Pinus elliottii*. Their study revealed that pta-miR949, pta-miR946a-3p, and pab-miR3710 target PITA39236 (HDS), PITA24638 (GPPS), and PITA12789 (ACAT), respectively. MiRNAs also target non-conserved proteins specific to particular terpenoid biosynthesis pathways. These studies focus on species-specific terpenoid synthesis, investigating key factors that distinguish one terpenoid from another by modulating the mRNA expression of downstream non-conserved proteins. Pani et al. [61] found that miR-5021 targets UDP-glucose iridoid glucosyltransferase in *C. roseus*, which is responsible for the glucosylation step in the biosynthetic pathway of the secondary metabolite iridoid in higher plants. Additionally, miR-5021 targets strictosidine synthase, an enzyme that catalyzes the condensation of tryptamine and the iridoid secologanin to produce strictosidine, the universal precursor of terpenoid indole alkaloids (TIAs).

### 3.3. Epigenetic Regulatory Mechanisms in Terpene Biosynthesis

Epigenetic regulatory mechanisms play a crucial role in modulating terpenoid biosynthesis through processes such as DNA methylation, histone modification, and non-coding RNA regulation. MiRNA regulation is also recognized as a component of epigenetic control [62]. In *Populus × canescens* (gray poplar), diurnal variations in terpene emissions are closely linked to fluctuations in the expression of isoprene synthase (ISPS) genes. These fluctuations are regulated by circadian clock-associated elements, specifically CCA1/LHY, within the ISPS promoter region [63]. UV-B irradiation of indoor-grown spruce seedlings has been shown to induce epigenetic modifications, leading to selective terpene release [64]. Studies on *Thymus kotschyanus* have demonstrated that methyl jasmonate (MeJA) can induce DNA hypomethylation, leading to the upregulation of *CYP450* gene expression [65]. These studies reveal the relationships between epigenetic regulation and plant circadian rhythms, growth environmental factors, and environmental signal transduction, indicating the critical role of epigenetic regulatory mechanisms in the biosynthesis of plant terpenoids.

### 3.4. Environmental Factors and Phytohormones in Modulating Terpene Biosynthetic Pathways

Various environmental stimuli, e.g., light, temperature, and biotic stress, significantly altered the expression of genes involved in terpenoid production [66,67]. Exposure to UV light and elevated temperatures often enhance the biosynthesis of protective terpenoids [68]. UV-B radiation can activate the plant defense system and promote the biosynthesis of secondary metabolites such as terpenes and phenolics [69]. Meanwhile, UV-A and visible light (400–700 nm) can act synergistically to facilitate the production of UV-B-absorbing compounds, thereby enhancing plant resistance to UV-B [70]. Phytohormones regulate terpenoid biosynthesis in response to environmental stress [71,72,73]. MeJA treatment increased terpene emissions in *Picea abies*, with linalool rising > 100-fold and sesquiterpenes > 30-fold, peaking during light periods [74]. Salicylic acid (SA) has been demonstrated to upregulate key enzymes in the terpenoid biosynthesis pathway, including FPPS in *Euphorbia pekinensis* [75] and SQS in licorice (*Glycyrrhiza* spp.) [76]. In *Michelia chapensis*, both JA and SA induced expression of the *MichHMGR* gene [71]. Under adverse conditions such as drought or salt stress, an imbalance occurs between the energy captured by plants and their capacity to process it. Plants employ various mechanisms to prevent and dissipate this excess excitation energy. Some of these mechanisms involving the upregulation of terpenoid levels occur concomitantly with increased SA levels [77,78].

Interestingly, environmental factors may modulate miRNA expression, thereby indirectly regulating downstream targets. For instance, in the leaves of *Camellia sinensis* (commonly used for tea production), terpenoids are a critical indicator for the tea’s aroma. Studies have shown that miRNAs in *C. sinensis* may be regulated by photoreceptors, in turn, which exert a negative regulatory effect on their target genes, ultimately modulating terpenoid biosynthesis [79]. This intricate regulatory network highlights the complexity and precision with which plants respond to environmental cues, optimizing the production of secondary metabolites such as terpenoids.

## 4. Novel Terpenoid Resources

Certain diterpenes possess semi-volatility and serve as phytoanticipins. They can either be synthesized as a response to pests, pathogens, or elicitors or they are constitutively produced during the normal growth process of plants. Many non-volatile terpenes are secreted by plant roots, acting as a primary defense mechanism and shaping rhizosphere communities [80,81]. In the contrast, plant-released volatile organic compounds (VOCs) facilitate plant-to-plant communication, and most of the VOCs are terpenes [82]. VOCs help plants lure pollinators, aiding in pollination [83].The formation of tree heartwood often involves the accumulation of secondary metabolites, which contribute to its distinctive darker color and provide protection against decay and insect infestation [84]. In the sandalwood heartwood, sandalwood oil (including α-santalol, β-santalol, epi-β-santalol, and α-exo-bergamotol) accumulates driven by the activity of SaSSY, a terpene synthase enzyme. These terpenoids provide resistance to decay and contribute to the long-lasting fragrance of sandalwood, enhancing its value as a high-quality timber [85]. Insects exposed to essential oil terpenes exhibited reduced respiratory rates, as well as avoidance or decreased movement on surfaces treated with these compounds. Terpenes such as eugenol, caryophyllene oxide, α-pinene, α-humulene, and α-phellandrene were found to be toxic to *Sitophilus granarius* and capable of deterring its predatory behavior [86].

Although terpenoids are primarily recognized for their roles in plant physiology and ecology, their economic and medicinal potential is increasingly being acknowledged [3,87]. The diversity of terpenoids arises from variations in enzymes, substrate specificities, and secondary modifications of terpene synthases. Terpenoid compounds exhibit diverse therapeutic applications. For instance, artemisinin, a potent antimalarial compound biosynthesized in *Artemisia annua*, has also demonstrated efficacy in cancer treatment [88,89]. Similarly, tanshinones, a class of abietane diterpenes derived from *Salvia miltiorrhiza*, are widely used in managing inflammatory conditions and show promising potential in treating neurological disorders [90,91,92,93]. Additionally, other terpenoids, such as oleanolic acid [94], myrcene [95], limonene [96], and linalool [97], have also been recognized for their significant pharmacological benefits.

Known for their aromatic properties and health benefits, terpenes are widely used in cosmetics and food products. Many essential oils (EOs) are rich in terpenoids, including sandalwood EO, as well as limonene (derived from citrus peels) [98], linalool (a key compound in producing household products, cosmetics, and fragrance chemicals such as geraniol, nerol, citral, and their derivatives) [99], and eucalyptus essential oils (EEOs) [100]. Due to their distinctive aromas, these terpenoid-rich EOs are widely used in personal care products such as hair tonics, soaps, and toothpaste. Limonene, linalool, and eucalyptol exhibit antioxidant properties, making them valuable additives in cosmetics to protect the skin against UV-induced damage [3]. As natural bioactive compounds, terpenoids can also be utilized to produce functional foods with health benefits. For instance, consuming terpenoid-enriched functional foods may help supplement type 2 diabetes treatment, potentially reducing the side effects associated with single-drug therapies [101]. Moreover, terpenoids serve as effective food preservatives. Terpenoids like erpenoids act as natural preservatives in the food industry due to their antimicrobial properties. Terpineol isomers (α-terpineol, terpinen-4-ol, and δ-terpineol) exhibit strong inhibitory effects against Gram-negative bacteria by disrupting membrane permeability and causing cell damage [102].

Interest in these compounds has grown due to the increasing demand for natural products and heightened regulatory scrutiny. Synthetic biology provides innovative solutions to enhance their production. For example, geraniol, an acyclic monoterpenoid alcohol and a major component of essential oils such as rose oil and citronella oil, has been produced at high levels in *Escherichia coli* through the recombinant overexpression of specific enzymes [33,103]. Similarly, antalene and santalol have been synthesized in vitro using engineered *Saccharomyces cerevisiae* with overexpressed enzymes [104]. Numerous high-performance engineered systems for plant terpene synthesis have been developed. Ignea et al. [105] engineered Saccharomyces cerevisiae by converting the yeast farnesyl diphosphate synthase (Erg20p) into a geranyl diphosphate synthase, establishing a dedicated platform for terpenoid biosynthesis. Basallo et al. [106] engineered rice lines with an MVA biosynthetic pathway, expressing three alternative versions of the MVA pathway in plastids.

## 5. Conclusions

Plant terpenoids are gaining prominence due to their broad application prospects. Recent advances have revealed their innovative uses in pharmaceuticals, food additives, and chemical industries, demonstrating significant market potential. Several critical knowledge gaps remain, particularly regarding the integration of different regulatory layers, including transcriptional, post-transcriptional, and epigenetic mechanisms. The precise roles of many newly identified TFs and miRNAs across various plant species, as well as their interactions with environmental factors, are still poorly understood. Moreover, the spatial and temporal dynamics of these regulatory networks during plant development and in response to biotic and abiotic stresses remain largely unexplored. These limitations underscore the need for more comprehensive systems biology approaches and functional validation studies to unravel the complex regulatory mechanisms governing terpenoid biosynthesis. Such investigations are crucial for deepening our understanding of plant metabolism and will provide valuable insights for metabolic engineering strategies aimed at optimizing terpenoid production for industrial and pharmaceutical applications. Based on that knowledge, future research should be directed toward the mechanisms governing terpenoid biosynthesis dependent on environmental change and molecular breeding.

## Figures and Tables

**Figure 1 plants-14-01428-f001:**
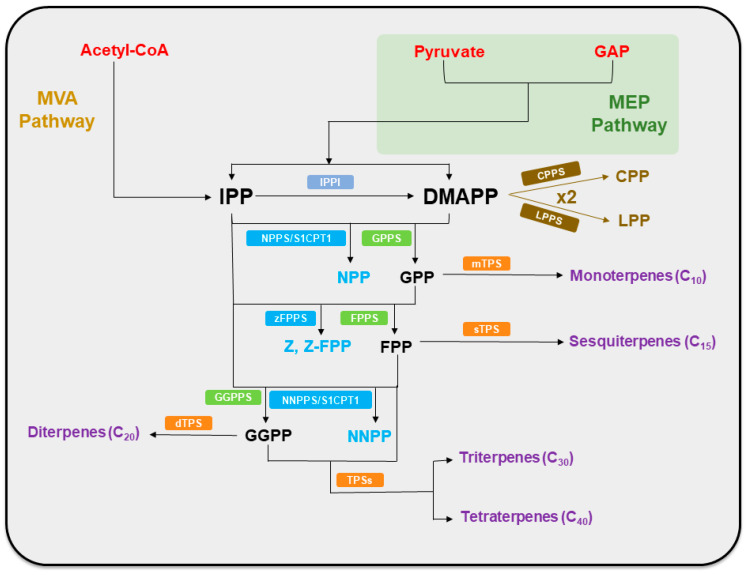
Schematic representation of plant terpenoid and terpene biosynthesis. The green areas represent plastids. Abbreviations: GAP, D-glyceraldehyde-3-phosphate; MVA pathway, mevalonate pathway; MEP pathway, 2C-methyl-D-erythritol-4-phosphate pathway; IPP, isopentenyl diphosphate; IPPI, isopentenyl diphosphate isomerase; DMAPP, dimethylallyl diphosphate; CPPS, chrysanthemyl diphosphate synthase; CPP, chrysanthemyl diphosphate; LPPS, lavandulyl diphosphate synthase; LPP, lavandulyl diphosphate; NPPS/S1CPT1, nerolidol diphosphate synthase; GPPS, geranyl diphosphate synthase; NPP, nerolidol diphosphate; GPP, geranyl diphosphate; mTPS, monoterpene synthase; zFPPS, Z, Z-FPP synthases; FPPS, farnesyl diphosphate synthase; Z, Z-FPP, Z, Z-farnesyl diphosphate; FPP, farnesyl diphosphate; sTPS, sesquiterpene synthase; GGPPS, geranylgeranyl diphosphate synthase; NNPPS/S1CPT2, nerylneryl diphosphate synthase; dTPS, diterpene synthase; GGPP, geranylgeranyl diphosphate; NNPP, nerolidol diphosphate; TPSs, terpene synthases.

**Figure 2 plants-14-01428-f002:**
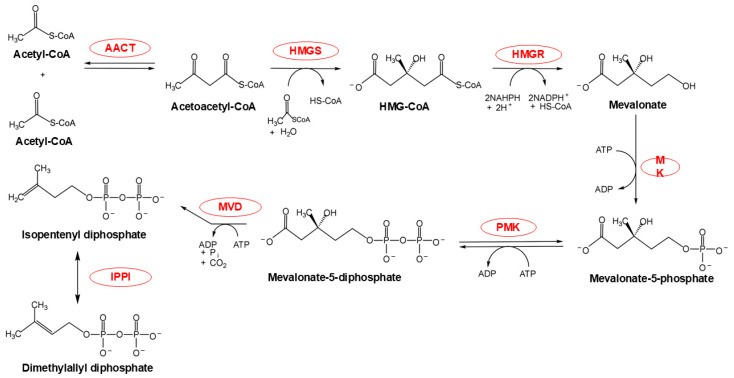
MVA pathway chemical flow. The red portion highlights the enzymes participating in this reaction. Abbreviations: AACT, acetoacetyl-CoA thiolase; HMGS, 3-hydroxy-3-methylglutaryl-CoA synthase; HMG-CoA, 3-hydroxy-3-methylglutaryl-CoA; HMGR, HMG-CoA reductase; MK, mevalonate kinase; PMK, phosphomevalonate kinase; MVD, mevalonate diphosphate decarboxylase; IPPI, isopentenyl diphosphate isomerase.

**Figure 3 plants-14-01428-f003:**
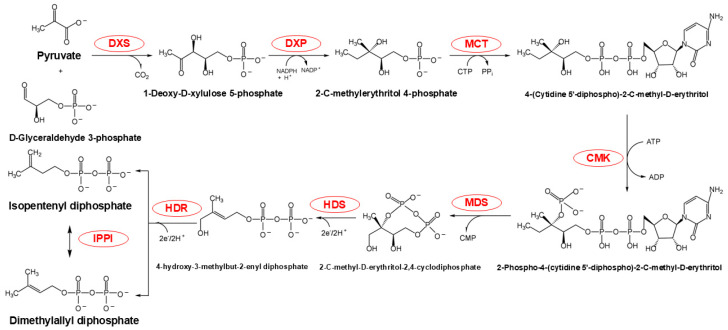
MEP pathway chemical flow. The red portion highlights the enzymes participating in this reaction. Abbreviations: DXS, 1-deoxy-D-xylulose-5-phosphate synthase; DXR, DXP reductoisomerase; MCT, 2-C-methyl-D-erythritol 4-phosphate cytidylyltransferase; CMK, 4-(cytidine 5-diphospho)-2-C-methyl-D-erythritol kinase; MDS, 2-C-methyl-D-erythritol-2,4-cyclodiphosphate synthase; HDS, 4-hydroxy-3-methylbut-2-enyl diphosphate synthase; HDR, 4-hydroxy-3-methylbut-2-enyl diphosphate reductase; IPPI, isopentenyl diphosphate isomerase.

**Figure 4 plants-14-01428-f004:**
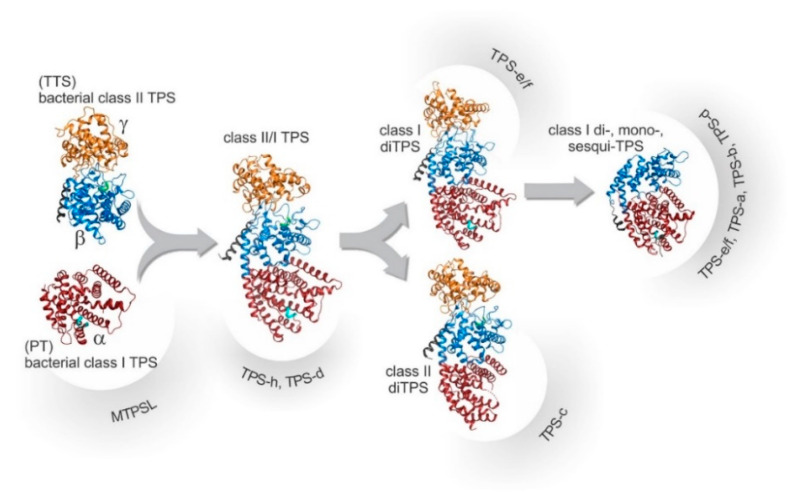
Evolutionary relationships among TPS based on protein structures. The domain color marks the γ-domain (orange), the β-domain (blue), the α-domain (red) as well as the conserved DxDD (green) and DDx2D (cyan) motifs.

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
