# Peer review of "Advances in the Biosynthesis of Plant Terpenoids: Models, Mechanisms, and Applications"

_plants, 2025, doi:10.3390/plants14101428_

Round 1
Reviewer 1 Report
Comments and Suggestions for Authors
The authors presented the review on the recent advances in the biosynthesis of plant terpenoids. However, there are many deficiencies in this manuscript. Also, the content of this manuscript is insufficient and incomplete. The manuscript does not meet the standards of the journal, Plants. Therefore, this manuscript is not recommended for publication. Listed below are my specific comments.
- This manuscript is incomplete as a "review." The authors should state the time period of the literature covered in this manuscript. Have the authors covered all previously published articles on the topic addressed in this manuscript? The authors should state these points.
- The authors should mention whether similar reviews have already been published on the topic covered in this manuscript.
- The authors should mention how they searched for literature on the topic covered in this manuscript, e.g., search engines and keywords.
- Line 28; The "Introduction" is redundant. The authors should provide a concise summary of terpenoids and their biosynthesis. The structural formulae of the compounds that appear in the text should also be included in the manuscript. The authors should clearly state the purpose of this review in the manuscript.
- The contents of Figures 1-5 are not mentioned in the text. The figures serve as a supplement to the topics explained in the text. The authors should reconsider this point. Also, the legends on the figures are redundant and should be kept concise.
- Many reviews on terpenoid biosynthesis have been published. The authors should differentiate this review from previous reviews. In this review, the authors should summarize the latest knowledge on the biosynthesis of mono-, sesqui-, di-, tri- and tetratepenoids, which will be more useful for the readers of this journal.
Author Response
Comments 1: This manuscript is incomplete as a "review." The authors should state the time period of the literature covered in this manuscript. Have the authors covered all previously published articles on the topic addressed in this manuscript? The authors should state these points.
Response 1: We appreciate the reviewer’s valuable feedback regarding the scope and completeness of our review. Below, we clarify the time frame and coverage of the literature included in this manuscript:
This article provides a systematic review of the research progress on the biosynthesis of plant terpenoids. It begins with an overview of the fundamental biosynthetic pathways of terpenoids, primarily based on classical studies from the 1990s to the early 2000s (e.g., Tamogani et al., 1993; Wang et al., 2000; Burt et al., 2004, etc.). With advancements in research technologies, studies in the 2010s began to focus on key metabolic steps and the mechanistic roles of related enzymes in terpenoid synthesis (e.g., Sangari et al., 2010; Falara et al., 2011; Vranová et al., 2013, etc.). Entering the 2020s, research has further focused on molecular regulation, with recent findings uncovering intricate regulatory mechanisms, including transcription factor modulation, epigenetic modifications, and environmental response mechanisms (e.g., Wang et al., 2024; Tu et al., 2024; Li et al., 2025, etc.). As well as the applied development of novel plant terpenoids(e.g., Lowe et al, 2021; Hu et al., 2024; Câmara et al., 2024, etc). This review particularly highlights these cutting-edge advances, aiming to serve as a reference for in-depth research in the field of terpenoid biosynthesis.
Comments 2: The authors should mention whether similar reviews have already been published on the topic covered in this manuscript.
R 2: We sincerely appreciate the reviewers’ valuable comments regarding the distinctions between our manuscript and existing reviews. Below, we outline the unique contributions of our work:
In recent reviews on plant terpenoids, several studies have focused on specific aspects. The 2023 review by Li et al. [1] provided a concise overview of terpenoid biosynthetic pathways and associated genetic regulation, with particular emphasis on the ecological functions of homoterpenes DMNT and TMTT and their potential applications in biological pest control. Demurtas et al. [2], in 2023, systematically examined advances in our understanding of terpenoid transport mechanisms within plant systems. More recently, Siddiqui et al. [3], in 2024 conducted a comprehensive analysis of terpenoids in plant essential oils, organizing their discussion by structural classification (monoterpenes, sesquiterpenes, diterpenes, and triterpenes) while detailing chemical properties, biological functions, and extraction methodologies. These three representative reviews collectively demonstrate that current terpenoid research has primarily addressed specialized topics within the field. In contrast, our work implements a multidimensional integrative approach to: (1) perform systematic comparisons of biosynthetic pathways and regulatory mechanisms across terpenoid subclasses; (2) introduce detailed classification and functional analysis of various synthase subtypes; and (3) provide novel insights into the molecular basis underlying terpenoid structural diversity - offering researchers a more holistic conceptual framework.
When compared with other comprehensive reviews such as Singh et al. [4] (2024), our distinct contributions include: (1) original phylogenetic analysis of subtype-specific synthases; (2) up-to-date coverage of emerging cooperative transcriptional networks, especially the interplay between transcription factors, epigenetic regulation, and phytohormone signaling - an area receiving limited attention in previous reviews; and (3) a dedicated section examining cutting-edge applications of novel terpenoids in pharmaceuticals, food science, and industrial chemistry, including recent progress in synthetic biology approaches for terpenoid production.
This review bridges fundamental research and applied science, presenting both systematic theoretical integration and innovative perspectives on terpenoid utilization.
Comments 3: The authors should mention how they searched for literature on the topic covered in this manuscript, e.g., search engines and keywords.
R 3: We appreciate the reviewer's valuable suggestion regarding our literature search methodology. We provide a detailed description of our systematic approach as follow:
The literatures in this manuscript included peer-reviewed articles, seminal reviews, and high-impact studies from databases, such as PubMed and Web of Science. To ensure comprehensive coverage of the research field, we implemented a systematic literature search strategy. When investigating the fundamental processes of terpenoid biosynthesis, we employed a series of broad keyword combinations like "plant terpenoids", "secondary metabolites", and "terpene biosynthesis", while consulting relevant chapters in authoritative plant physiology monographs. Such approach enabled us to systematically organize the basic biochemical processes of plant terpenoid synthesis. As the research progressed, we have focused particularly on the diversity characteristics of key metabolic pathways and related enzymes. We adopted more precise search strategies Using specific metabolic pathways (e.g., MEP pathway; MVA pathway) as keywords; andconducting specialized searches for key enzyme families (e.g., IDSs; TPSs; CYP450s). Through cross-species comparative analysis, we systematically examined the evolutionary differences and functional differentiation of these metabolic nodes and enzyme classes, while integrating current research findings on the molecular mechanisms underlying these key variations.
For the emerging field of molecular regulation research, we adopted a recency-prioritized literature screening approach: First, we developed targeted keyword combinations corresponding to the three research directions established in Section 3 (transcriptional regulation, epigenetic regulation, and environmental response regulation). Then we chronologically sorted the search results, with particular emphasis on selecting research publications from the most recent five years (2020-2025). Finally, we conducted systematic comparisons of publication trends, research focus evolution, and major breakthrough developments across these research directions, especially focusing on post-2020 advancements. While we prioritize key breakthroughs, some niche or highly specialized studies may not be exhaustively cited due to space constraints. However, we have ensured representation of all significant trends that mentioned above.
Comments 4: Line 28; The "Introduction" is redundant. The authors should provide a concise summary of terpenoids and their biosynthesis. The structural formulae of the compounds that appear in the text should also be included in the manuscript. The authors should clearly state the purpose of this review in the manuscript.
R 4: We sincerely appreciate the reviewer’s suggestions for improving our manuscript. We have deleted redundant content and reorganized the introduction into three paragraphs with logical progression:
(1) Overview of the basic characteristics of plant secondary metabolism and terpenoid compounds.
(2) Analysis of the ecological significance and application value of terpenoid compounds.
(3) Elaboration on the core objectives and research framework of this review.
The picture of structural formulas of compounds in the MEP and MVA pathways are illustrated in Figures 2 and 3. However, unconventional terpene precursors and specific terpenoids feature highly complex structures that cannot be adequately depicted using atomic-count-based shorthand notations. We would appreciate the reviewers' guidance on whether consolidating the discussed compound structures into a summary table might enhance readability for readers?
Comments 5: The contents of Figures 1-5 are not mentioned in the text. The figures serve as a supplement to the topics explained in the text. The authors should reconsider this point. Also, the legends on the figures are redundant and should be kept concise.
R 5: We have carefully revised the figures as follows:
- Regarding Figure 1, In light of revisions made to the Introduction section, particularly the streamlining of content regarding terpenoid applications (as suggested), we have determined that Figure 1 is no longer aligned with the current text and have consequently removed it. The discussion on the development and applications of terpenoids has been reorganized as a new section - Section 4, see lines 327.
- For Figure 2 (Figure 1 in the revised version), We have remarked the relevant parts of the figure in the text, please refer to lines 74, 316 and 319.
- For Figure 3 and 4 (Figure 2 and 3 in the revised version), the explanations of two figures have been added in the revised manuscript, see lines 89 and 109.
- For Figure 5 (Figure 4 in the revised version), we have streamlined its caption to improve clarity. Additionally, we have expanded the description of figure in the main text to provide more detailed explanations, see lines 202-206.
Comments 6: Many reviews on terpenoid biosynthesis have been published. The authors should differentiate this review from previous reviews. In this review, the authors should summarize the latest knowledge on the biosynthesis of mono-, sesqui-, di-, tri- and tetratepenoids, which will be more useful for the readers of this journal.
R 6: We sincerely appreciate your valuable suggestions once again regarding the novelty of our article, the response to this comment is identical to the solution for comments 2.
Reference
- Li, C.; Zha, W.; Li, W.; Wang, J.; You, A. Advances in the biosynthesis of terpenoids and their ecological functions in plant resistance. International journal of molecular sciences 2023, 24, 11561.
- Demurtas, O.C.; Nicolia, A.; Diretto, G. Terpenoid transport in plants: how far from the final picture? Plants 2023, 12, 634.
- Siddiqui, T.; Khan, M.U.; Sharma, V.; Gupta, K. Terpenoids in essential oils: Chemistry, classification, and potential impact on human health and industry. Phytomedicine plus 2024, 4, 100549.
- Singh, S.; Chhatwal, H.; Pandey, A. Deciphering the Complexity of Terpenoid Biosynthesis and Its Multi-level Regulatory Mechanism in Plants. Journal of Plant Growth Regulation 2024, 43, 3320-3336.
Reviewer 2 Report
Comments and Suggestions for Authors
The manuscript provides detailed information on biosynthesis of plant terpenoids, it can be accepted for publication after correcting few minor shortcomings:
- italics is missing in some scientific names on plant species and Latin names (lines 82, 85, 96)
- the chapter “3.4 Environmental factors and phytohormones in modulating terpene biosynthetic pathways” could be elaborated in more detail, e.g. findings how the environmental factors affect the biosynthesis of terpenes and how it is reflected
- title of this manuscript include applications of biosynthesis of plant terpenoids, however it is not sufficiently described and elaborated in the text
Author Response
Comments 1: Italics is missing in some scientific names on plant species and Latin names (lines 82, 85, 96)
R 1: We have carefully reviewed the manuscript and corrected the formatting of all scientific names by italicizing them, including those in lines 82, 85, and 96.
Comments 2: The chapter “3.4 Environmental factors and phytohormones in modulating terpene biosynthetic pathways” could be elaborated in more detail, e.g. findings how the environmental factors affect the biosynthesis of terpenes and how it is reflected
R 2: We sincerely appreciate the reviewer’s valuable suggestion. We have expanded Section 3.4 with more detailed discussions, please see lines 301-326.
Comments 3: Title of this manuscript include applications of biosynthesis of plant terpenoids, however it is not sufficiently described and elaborated in the text.
R 3: We have added a dedicated new section (Section 4: " Novel terpenoid resources ") that comprehensively discusses. These changes appear in lines 327-387 of the revised manuscript.
Reviewer 3 Report
Comments and Suggestions for Authors
Review
Advances in the Biosynthesis of Plant Terpenoids: Models, Mechanisms, and Applications
A brief summary
An interestingly written review, taking into account key themes. However, I have a few comments.
Broad comments
1. The authors set themselves the task of presenting the current state of knowledge on the synthesis of terpenoids, compounds that are extremely important both for the functioning and ecology of plants and for the huge range of possible applications.
2. The work gives the impression of being well thought out and written and seeks to answer a number of important questions and fill in the gaps so far and draws attention to issues that have not been fully explained.
3. The compilation is developed in a comprehensive manner and addresses a number of issues relevant to understanding the synthesis of terpenoids by plant organisms in interaction with the environment.
4. A few comments:
a. The introduction seems at first to be too detailed, but compared to the issues raised in the following chapters, this impression tends to pass.
b. Keywords are the right place to put phrases that are not included in the title but characterise the work. Here, I suggest removing ‘terpenoids’ and ‘biosynthesis’ and replacing them with, for example, ‘mevalonate’ and ‘methylerythritol phosphate’.
5. The summary is an identification of areas that have not yet been clarified and an indication of directions for exploration, rather than a summary of the issues presented in the work.
6. I suggest looking more closely at the literature review. For example, the items listed in lines 244 and 246 are likely to have been misquoted, as they refer to the authors' first names rather than their surnames.
7. Additional comments.
a. Line 155 In this line there is text that should rather not be there.
Author Response
Comments 1: The introduction seems at first to be too detailed, but compared to the issues raised in the following chapters, this impression tends to pass.
R 1: Our intention was to provide sufficient background to contextualize the study. However, we acknowledge that the initial impression of density might affect readability. To address this, we have streamlined some parts of the introduction. Furthermore, we have reorganized the logical structure of the introduction section and added a clear statement of this review's objectives in the concluding paragraph.
Comments 2: Keywords are the right place to put phrases that are not included in the title but characterise the work. Here, I suggest removing ‘terpenoids’ and ‘biosynthesis’ and replacing them with, for example, ‘mevalonate’ and ‘methylerythritol phosphate’.
R 2: We have removed the redundant terms 'terpenoids' and 'biosynthesis' and added more specific keywords that have mentioned, including 'mevalonate pathway', 'methylerythritol phosphate pathway '…, see lines 27, 28.
Comments 3: The summary is an identification of areas that have not yet been clarified and an indication of directions for exploration, rather than a summary of the issues presented in the work.
R 3: We have revised and reorganized the abstract and conclusion section, highlighting the unresolved research areas in the development of novel terpenoid compounds as well as in the study of molecular regulatory mechanisms governing terpenoid biosynthesis.
Comments 4: I suggest looking more closely at the literature review. For example, the items listed in lines 244 and 246 are likely to have been misquoted, as they refer to the authors' first names rather than their surnames.
R 4: Sorry for the mistakes. We have conducted a thorough re-examination of all cited references throughout the manuscript, specifically corrected the citations in lines 244 and 246 to properly reflect authors’ surnames. We also verified the accuracy of all other citations in the reference list, to ensure consistency with the journal’s citation style. We thank the reviewer for bringing this important detail to our attention, as proper attribution is fundamental to academic integrity.
Comments 5: Line 155 In this line there is text that should rather not be there.
R 5: We have removed the unintended text from line 155 and carefully reviewed the surrounding content to ensure proper flow and coherence.
Round 2
Reviewer 1 Report
Comments and Suggestions for Authors
This revised manuscript has been modified according to the reviewer’s comments. It is acceptable for publication.